# Effect of Brewing Conditions on Antioxidant Properties of Ginkgo biloba Leaves Infusion

**DOI:** 10.3390/antiox12071455

**Published:** 2023-07-19

**Authors:** Patrycja Biernacka, Katarzyna Felisiak, Iwona Adamska, Marek Śnieg, Cezary Podsiadło

**Affiliations:** 1Department of Food Science and Technology, Faculty of Food Science and Fisheries, West Pomeranian University of Technology in Szczecin, 70-310 Szczecin, Poland; 2Department of Agroengineering, Faculty of Environmental Management and Agriculture, West Pomeranian University of Technology in Szczecin, 70-310 Szczecin, Poland

**Keywords:** *Gingko biloba*, liquid, antioxidant activity, migration of micro- and macroelements, sensory evaluation

## Abstract

Due to the growing awareness of the importance of healthy eating in society, there is an increasing interest in the use of herbs and low-processed, natural products. *Ginkgo biloba* is a raw material with a high pro-health potential, which is related to the high content of antioxidant compounds. The aim of the study was to determine the relationship between the antioxidant activity of *Ginkgo biloba* leaf infusions and the weighted amount of leaves and brewing time. In addition, a sensory analysis of the infusions obtained was carried out. The innovation is to determine the migration of micro- and macroelements to the infusion prepared from *Ginkgo biloba* depending on the leaves’ weight used and the brewing time. The research showed the dependence of the antioxidant activity of the infusions and the migration of microelements on the size of the dried material and the brewing time. In the publication, the main factors influencing the quality of infusions were analysed, their mutual correlations were determined, and combinations showing the highest antioxidant activity and, at the same time, the highest sensory acceptability were selected.

## 1. Introduction

Nowadays, people are more susceptible to diseases caused by an overly active lifestyle, professional ambitions and the related increasing stress level, and, at the same time, decreasing physical activity and the possibility of regeneration (adequate length of sleep). For this reason, the importance of disease prevention and superfoods is growing [1]. Products containing natural ingredients with health-promoting properties that support the body from the “inside” are becoming more and more popular.

*Ginkgo biloba* leaves and seeds, due to their health-promoting properties, are often used in medicine [2]. Dried leaves are used in the treatment of diseases of the central nervous system (i.e., Alzheimer’s, Parkinson’s), circulatory system (i.e., hypertension), dyslipidaemia, hyperglycaemia and obesity, and supplements based on ginkgo leaves are taken to improve memory and concentration [3,4,5,6]. Due to their high antioxidant and free radical scavenging properties, they are also credited with anti-ageing properties and improving the condition of hair and skin, e.g., when fighting cellulite [7,8].

The main components of ginkgo leaves are terpenoids, which include ten types of ginkgolides and bilobalide and its isomers. Flavonoids also play a key role in the health-promoting effect [9]. In addition, both *Ginkgo biloba* nuts and leaves contain a beneficial composition of minerals and vitamins; hence, they can be used to produce food rich in nutrients [10,11]. These properties make *Ginkgo biloba* a valuable raw material for the production of health-promoting products.

The aim of the publication is to determine the dependence of the antioxidant properties of infusions from dried *Ginkgo biloba* leaves on the weighted amount of leaves and brewing time. It was also important to determine the content and migration capacity of micro- and macroelements in dried *Ginkgo biloba* leaves to infusions during the brewing process.

The results of the conducted research will facilitate the development of food products with the use of *Ginkgo biloba*. This may provide the necessary information on the possibility of obtaining infusions with the best antioxidant activity and sensory qualities. During the research, the ability to migrate micro- and macroelements from the leaves to the infusions during brewing was analysed, and a detailed sensory assessment was presented. The publication also includes the results of the chemical analysis of extracts from dried *Ginkgo biloba* leaves obtained by various extraction methods and compares them with each other. The work focused on the synergy between various components present in *Ginkgo biloba* and their impact on the antioxidant activity of the infusion. Liquid chromatography analysis was carried out, which allowed us to isolate and determine the percentages of some biologically active compounds present in selected infusions. The biplot analysis showed many dependencies that would be very difficult to show when comparing the results in tables. During the research, the influence of temperature, weight of dried *Ginkgo biloba* and brewing time on changes in individual components was demonstrated. This may be important for understanding the mechanisms of the antioxidant action of infusions. So far, such extended analyses have not been carried out for *Ginkgo biloba* infusions.

## 2. Materials and Methods

### 2.1. Materials

The research material was dried *Ginkgo biloba* leaves from China, purchased in the online store planteon.pl. Whole leaves and boiling drinking water were used to prepare infusions. All reagents used were analytical grade (Analab, Warszawa, Poland and Bytomska Wytwórnia, Chorzow, Poland). 2-2′-Azino-bis(3-ethylbenzothiazoline-6-sulfaic acid) diammonium salt (ABTS), 2,2-diphenyl-1-picrylhydrazyl (DPPH), 2,4,6-Tris(2-pyridyl)-s-triazine (TPTZ), 6-Hydroxy-2,5,7,8-tetramethylchromane-2-carboxylic acid (Trolox) and aluminium chloride (III) were purchased from Sigma, Neustadt, Germany.

### 2.2. Basic Composition of Dried Leaves

Samples were taken from the leaves of *G. biloba* (dried) for the analysis of the basic composition. The contents of dry matter, total nitrogen (Kjeldahl method, Kjeltec KT 200 apparatus, Labtec Line, FOSS), fat (Soxhlet method) and ash (dry mineralisation method) were determined in accordance with the Association of Official Analytical Chemists (AOAC) [12]. In addition, water extracts were prepared from the dried material using the shaking method (2 h, room temperature, laboratory shaker type WL-1) in two variants: using cold distilled water (LWC) and boiling distilled water (LWH) (5 g/50 mL of water) and methanol extracts (5 g/50 mL 80% methanol). The principle that was used to determine the content of carbohydrates is the difference between the dry weight of leaves and the sum of the masses of ash, fat and protein contained in it.

### 2.3. Infusions

Infusions from *G. biloba* leaves were prepared in 20 variants, differing in leaf weight and brewing time. The tests included weights of 1.25 g (X), 2.5 g (S), 5 g (M) and 7.5 g (L). The herbs were always poured with 250 mL of boiling drinking water. The brewing time was 5 min, 10 min, 15 min, 20 min and 60 min (Table 1). After a certain time, the leaves were separated from the samples. From the infusions cooled to room temperature (20 °C), methanol and water extracts were then prepared (10 mL of the infusion and 40 mL of pure methanol or water, respectively).

### 2.4. Determination of pH

The pH was determined using a digital pH meter (PL-700PVS; ChemLand, Stargard, Poland).

### 2.5. Determination of Apparent and Real Extract

The soluble dry matter (so-called extract) of infusion was measured using a refractometer (portable refractometer).

### 2.6. Determination of Acidity

The acidity was determined by the titration method using 0.1 M NaOH in the presence of phenolphthalein (1% ethanol solution), according to Krełowska-Kułas (1993) [13]. The results are expressed as the amount of acetic acid in g/L.

### 2.7. Chemical and Physical Analyses

The total content of polyphenols was determined in water and methanolic fractions according to the method described by Turkmen et al. (2005) [14]. Briefly, 1 mL of the diluted sample, 5 mL of 10% Folina–Ciocalteau reagent, and 5 min later, 4 mL of 7.5% Na_2_CO_3_ were added. After 2 h incubation in the dark, absorption was measured at 750 nm (Helios Gamma spectrophotometer, Thermo Spectronic, Horsham, UK). Results were expressed as mg of gallic acid per 1 mL of infusion.

Free DPPH˙ radicals scavenging ability (DPPH) was determined according to the method described by Tang et al. (2002) [15]. Briefly, 4 mL of methanolic extract and 1 mL of 0.2 μM DPPH˙ in methanol were added. The samples were then shaken and left in the dark for 30 min, and the absorbance was measured at 517 nm. Free radical scavenging ability was expressed as μM Trolox/mL based on a decrease in absorbance relative to the control (%) using the Trolox standard curve.

Trolox Equivalent Antioxidant Capacity (TEAC) was determined according to the method described by Re et al. (1999) [16]. It is proportional to the degree of reduction of the ABTS + radical by antioxidants contained in the prepared methanol and water extracts. A total of 4 mL of the ABTS solution (7 µM solution activated by K_2_S_2_O_8_ and diluted to absorbance 0.700 ± 0.020 directly before analyses) was mixed with 40 μL of the tested extract and left in a dark place for 30 min. After this time, the absorbance was measured at a wavelength of 734 nm. Total antioxidant activity was expressed as µM Trolox/mL based on the standard curve.

Ferric reducing antioxidant power (FRAP) (the ability to reduce Fe^3+^ ions to Fe^2+^) was carried out in water and methanol extracts according to the method described by Benzi and Strain (1996) [17]. A total of 3 mL of working solution (1A:1B:10C, where A is a solution of 0.01 M TPTZ in 0.04 M HCL, B is 0.02 M FeCl_3_, and C is 0.3 M acetate buffer with pH 3.6), previously heated at 37 °C for 30 min, was added to 100 μL of the sample. Samples were incubated for 30 min at room temperature in the dark, and then the absorbance at 593 nm was measured. Antioxidant activity was expressed as Trolox equivalents [µM Trolox/mL] from a standard curve.

Total Flavonoid content (TFC) was determined according to the method described by Shraim and colleagues (2021) [18]. A total of 0.5 mL infusion was dissolved with 1.5 mL of methanol in the test tube, and a reagent consisting of 0.1 mL of 10% AlCl_3_, 0.1 mL of 1M Sodium Acetate and 2.8 mL of distilled water was added. The sample was mixed and incubated for 30 min at room temperature. The absorption of the solution was measured on a UV–Vis spectrophotometer at 415 nm using a blank, where infusion was replaced with distilled water. Six samples differing in weight (2.5 g and 5 g) and brewing time (10 min, 15 min and 60 min) were selected for the study of infusions. The total flavonoid content was expressed as equivalents of quercetin [µg QE/mL].

### 2.8. Determination of Active Compounds by Liquid Chromatography (HPLC)

Flavonoids and polyphenolic acids from infusions were determined in six samples differing in weight (2.5 g and 5 g) and brewing time (10 min, 15 min and 60 min) by liquid chromatography (HPLC). Agilent 1260 Infinity II liquid chromatograph coupled with a PDA detector was used for the analysis. Separation was carried out on a 250 × 4.6 mm Nucleosil 120-5 C18 reverse phase column at ambient temperature. The mobile phase consisted of acetonitrile (solvent A) and water with 5% acetic acid (solvent B). The flow rate was maintained at 0.5 mL/min. The gradient program was 15% A/85% B 0–12 min, then linearly changed to 0% A/100% B at 30 min, followed by a change to 85% A and 15% B at 50 min and the pattern of 85% A and 15% B held for another 10 min. The total analysis time was 60 min. The injection volume was 20 µL, and peaks were monitored at several wavelengths: 270, 280, 290 and 325. Samples were filtered through a 0.45 µm membrane filter prior to injection [19]. Peaks of identified compounds were integrated, and the percentage shares of their areas in the chromatogram were determined.

### 2.9. Sensory Assessment

The sensory evaluation of the infusions was carried out using the hedonic method. The sensory evaluation involved five trained panellists who had no previous or present history of taste or smell disorders. Samples were coded (three-digit codes) and randomly presented to avoid presentation order bias. Assessment of colour, smell, flavour and overall desirability was performed by scaling using a 5-point scale (rating 1 meant “undesirable” and 5—“very desirable”). Sensory profiling of the flavour and smell of the infusions was also carried out using a 9-point scale (1—“imperceptible” and 9—“very perceptible”). The descriptors adopted were: tea, *Ginkgo biloba* tea, sour, bitter, astringent and sweet. The results of the profile analysis are presented in the form of a polar plot. The test was carried out in accordance with PN-ISO 3972 and ISO 53103:1996 [20,21].

### 2.10. Analysis of Micro- and Macroelements

The analysis of selected elements (Pb, Ni, Cu, Cd, Cr, Fe, Na, Mg, K, Ca, Mn) was performed with an atomic absorption spectrometer (Thermo Fisher Scientific iCE 3000 Series, Waltham, MA, USA). In the case of dried leaves, the samples were mineralised and then tested in various dilutions. Six samples differing in weight (2.5 g and 5 g) and brewing time (10 min, 15 min and 60 min) were selected for the study of infusions. The infusions were brewed with distilled water and filtered, and then a series of dilutions were prepared from them. 

### 2.11. Statistical Analysis

All chemical determinations were performed in triplicate and presented as mean values and standard deviations. Statistical analysis was based on a one-way analysis of variance (ANOVA). Homogeneous groups were created by Tukey’s test for *p* ≤ 0.05. Principal Component Analysis (PCA) and cluster analysis were also performed. Using PCA, correlations between the acidity of the infusions, soluble dry matter, the total content of polyphenols and the antioxidant activities determined by the FRAP, TEAC and DPPH were determined. On the unit circle, correlations between the active variables were marked [content of total polyphenols of the water fraction (Polyphenols W) and methanol fraction (Polyphenols M), FRAP of the water fraction (FRAP W), FRAP of the methanol fraction (FRAP M), TEAC of the water fraction (TEAC W), TEAC of the methanol fraction (TEAC M) and DPPH] and additional variables (acidity and extract content). In addition, a cluster analysis was carried out using the method of hierarchical cluster agglomeration in order to group the analysed indicators into main groups indicating the greatest affinity. The data was statistically analysed using Statistica 13.0 StatSoft Inc. data analysis software [22].

## 3. Results and Discussion

### 3.1. Basic Characteristics of Dried Ginkgo biloba Leaves

The dry weight of *G. biloba* leaves accounts for over 92% of the weight. The largest share has carbohydrates, which account for over 75%. The tested raw material contains about 12% of protein, 5% of fat and about 1% of ash (Table 2). The values obtained in our own research are definitely different from the values obtained in the study of ginkgo leaves growing in Nigeria: in our own research, the share of fat and protein was found to be almost half higher [23]. Probably, this difference may result from different cultivation conditions (soil richness in minerals) and the season in which the leaves were collected [24].

The highest total content of polyphenols is found in the methanol extract (169.5 mg GAE/g). This value is almost twice as high as the content of these substances in water extracts. However, no statistical difference is found between the total content of polyphenols in water extracts obtained by different extraction methods (cold and hot) (Table 3).

During the research, it was found that the antioxidant activity determined by the FRAP and TEAC methods was higher in methanol extracts than in water extracts. In addition, for both methods (FRAP and TEAC), there are significant differences in activity between the two types of water extracts: in the FRAP analysis, the total antioxidant activity is higher in the cold extract (LWC) and in the case of TEAC—in the hot extraction (LWH). The free radical scavenging capacity of DPPH is 10.852 µg TE/g. This value is five times higher than the results obtained by Sati (2013) [25] when analysing the activity of *G. biloba* leaves from different areas of India. Similar results were obtained in the study by Naliniwata et al. (2018) [26], where the activity of green tea leaves (*Camellia sinensis* L. Kuntze) was determined. The content of polyphenols in traditional tea commercially available in Argentina was almost eight times lower than the values of *G. biloba* in our own research [27].

The results confirm that *Ginkgo biloba* leaves have much higher antioxidant activity than infusions due to the presence of hydrophobic compounds [28].

### 3.2. Basic Composition and Antioxidant Activity of Infusions from Dried Ginkgo biloba Leaves

During the research, a decrease in the pH of the infusion was found with an increase in the weight of dried *G. biloba* leaves and with the extension of the brewing time. An infusion prepared from the smallest weight of leaves for the shortest time shows the highest pH value (sample X5; pH value 7.7) and is characterised by the slowest pH decrease to 6.76. The pH value of the initial samples decreases with the increasing weight, which contributes to the largest decrease in pH in the L samples—the initial pH is 6.48, while the final pH is 4.41 (Figure 1). The lowering of the pH is mainly due to the increasing amount of water-soluble substances, i.e., catechins, transferred from the leaves to the infusion [29].

The total content of polyphenols increases with increasing brewing time and with increasing weight (Table 4). The highest value is found in the L60 samples (1.098 mg GEA/mL in the methanol fraction and 1.317 mg GEA/mL in the water fraction), although a high content is also found in the M15 test (0.877 and 0.933 mg GEA/mL). The lowest content of polyphenols is found in sample X5 (0.315 mg GEA/mL in the methanol fraction and 0.169 mg GEA/mL in the water fraction). Polyphenols are a group with a very high biologically active potential; hence, they have an antioxidant effect. The content of these compounds, however, varies not only in individual plant species but can also greatly vary depending on the degree of leaf maturity, leaf harvesting season or genetic features of the plant [30].

A similar dependence is observed during the analysis of antioxidant activity using the TEAC method: the highest activity is found in samples L and M. The highest values are found in the methanol fractions of samples, L5 (1.573 µg TE/mL) and L60 (1.374 µg TE/mL), M15 (1.334 µg TE/mL), and in the water fractions of samples, L60 (2.162 µg TE/mL), L20 (1.990 µg TE/mL), L5 (1.755 µg TE/mL) and M15 (1.441 µg TE/mL). Different results are obtained using the DPPH method. In this case, the highest activity is found in sample S15 (0.300 µg TE/mL) and the lowest—X5 (0.020 µg TE/mL). Flavonoids, i.e., quercetin or kaempferol, can prevent the negative impact of free radicals on the body [31]. The ability to scavenge free radicals contributes to slower ageing, prevents the development of diseases of the central nervous system and the circulatory system and supports immune function [32]. According to the literature’s data, the antioxidant activity and the ability to scavenge free radicals of DPPH are significantly affected by the amount of phenolic compounds [33].

During the analysis of antioxidant activity using the FRAP method, an increase in the infusion activity in the methanol fraction is found with increasing brewing time, but this change is not correlated with the changing weight of *G. biloba* leaves. Although the samples with the highest mass (L) are characterised by the highest activity (L60: 0.530 µg TE/mL, L5: 0.448 µg TE/mL and L20: 0.411 µg TE/mL), high activity is also demonstrated in sample M15 (0.355 µg TE /mL). The lowest activity is found in sample X (with the smallest leaf weight). In the FRAP analysis of the water fraction, the highest activity is found in sample M15 (0.532 µg TE/mL) and the lowest in samples with the lowest leaf weight (X). The data show that the highest antioxidant activity is found in the infusion obtained with a brewing time of 15–20 min and dry leaf weight of 5 and 7.5 g (M and L, respectively).

The content of soluble dry matter (Table 5) in the infusion logarithmically increases until the 20th minute of brewing (0.57% at the culminating point in all samples) and then decreases to a level depending on the weight of the leaves (sample X: 0.0%; S: 0.0%; M: 0.2%; L: 0.5%).

A similar trend (logarithmic increase with increasing brewing time and increasing weight) is also observed for acidity (Figure 2). In test X, these values increase from 0.07 to 0.17 g/L, in test S from 0.09 to 0.22 g/L, in M—from 0.12 to 0.31 g/L and in L—from 0.17 to 0.41 g/L.

In the literature, no information was found on the content of the extract in infusions of dried *Ginkgo biloba* leaves and on their titratable acidity. Perreira et al. (2013) [28] only determined the content of fructose, glucose and sucrose. They found that fructose had the largest share and sucrose the smallest.

PCA show that the first two components account for 91.88% of the total variance (Figure 3A). The strongest correlation is found between the content of polyphenols in the water fraction (0.97) and methanol fraction (0.96), DPPH (0.95) and TEAC in the water fraction (0.95), while the lowest correlation is found for the soluble dry matter content (0.61) and acidity (0.16). In addition, vectors indicating acidity and soluble dry matter content indicate a strong negative correlation between these variables. The analysis allows the distribution of the trials in the quadrants of the two-factor case coordinate plot (Figure 3B). The trials that make up the one-piece sets in quadrant I are X5. This sample is distinguished by its low acidity. In quarter II, a separate group is formed by L60, L20 and S15, which are connected by high DPPH, TEAC of the water fraction and the content of polyphenols and FRAP of the methanol fraction. Quarter III samples contain a higher content of polyphenols and FRAP of water fraction, FRAP and TEAC of methanol fraction, and extract content than other cases. A separate group in this quarter are M15, L5 and M10. It is worth noting that in this quarter, the vast majority of samples belong to samples with a 5 g weight (M). Quadrant IV says about the lack of significant correlations with respect to antioxidant activity but shows strongly positively correlated variables represented by S20 and M10. It is worth noting that such parameters as the soluble dry matter content or the amount of polyphenols in the water fraction are the components determining the selection of the most desirable infusion. In this case, the M15 and L5 trials are at the forefront.

Indices of antioxidant activity in the hierarchical method of cluster agglomeration are grouped into 3 main groups. The total polyphenol content of the water fraction and methanol fraction is closest to the vertical axis, which indicates their highest correlation. This cluster group also includes DPPH and TEAC of the water fraction and FRAP of the methanol fraction. The second cluster is formed by the TEAC analysis of the methanol fraction. The distance of the FRAP activity of the water fraction from all clusters is the greatest, and it is considered an independent cluster. The large Euclidean distance indicates no similarities between these factors (Figure 4).

The value of the correlation coefficient between DPPH and the total polyphenol content of the methanol fraction (Table 6) in relation to all the tested samples is 0.94, which indicates the strongest linear relationship between free radical scavenging ability and polyphenols content (determination coefficient 0.87 and significance level *p* less than 0.05 indicate to the high significance of the correlation found).

### 3.3. Determination of Active Compounds by Liquid Chromatography (HPLC) and Total Flavonoid Content

*Ginkgo biloba* leaves contain various active compounds, the largest share of which are flavonoids and terpenes. These compounds are characterised by a strong and beneficial effect on the human body [34]. HPLC analysis of various infusions from dried *Ginkgo biloba* leaves allowed us to identify (Figure 5) and present percentages (Table 7) of five bioactive substances. In the samples, there are integrated chlorogenic acid, epicatechin, catechin, coumarin and apigenin. All of the identified compounds have a strong antioxidant [35,36], anticancer [37,38] and neuroprotective effect [39].

Table 7 shows the percentages share of isolated biologically active substances in the chromatogram. Among the samples, changes depending on the brewing time and weight can be noticed. It can be seen that the percentage of chlorogenic acid decomposes with increasing weight and brewing time. In the case of epicatechin and catechin, the percentages seem to be more stable, although their shares are higher in the longest brewing time compared with 10 min. On the other hand, the proportion of coumarin and apigenin decays with the brewing time for each sample.

The total content of flavonoids (TFC) in *Gingko biloba* infusion increases with weight and brewing time. The lowest content of flavonoids is obtained in S10, which contains 1.66 µg QE/mL, and the highest content in L60, which contains 5.97 µg QE/mL (Table 8). During the statistical analysis, it can be seen that the flavonoid content is statistically insignificant in the case of M60 and L10 samples. This proves that similar results can be obtained by increasing the weight without the need for such a long brewing time. The total flavonoid content in black tea, researched by Torre et al. (2021) [40], was 2.3 µg QE/mL, which corresponds to the S15 test (2.29 µg QE/mL). For this study, a 3 g sample weight and 15 min brewing time were used.

### 3.4. Sensory Assessment

In a 5-point scaling method, the colour, smell, flavour and overall acceptability of the infusions are assessed (Figure 6). The colour of the tested samples, in most cases, obtains the maximum number of points—5.00 points (X5, X10, X60, S5, S10, S60, M5, M20, M60, L5, L60). The test that is characterised by the least attractive colour is the X15 samples (2.00 points). The fragrance that scored best is X12 and S10 with a score of 5.00 points, and the lowest in X60 (1.75 points), X15 (2.25 points), L20 (2.63 points), S60 (2.75 points) and X20 (2.75 points). The most decisive distinguishing feature in the overall assessment of the acceptability is the flavour; it shows statistically significant differences (*p* > 0.05) when comparing individual samples. The highest scores for flavour and general desirability are obtained for sample M15 (4.75 and 4.50 points, respectively), while the lowest scores are for infusions obtained from the largest weight of dried leaves [L5, L10, L60 (1.00 points), L15 (2.50 points) and L20 (1.50 points)].

In the case of samples in which the brewing time was in the range of 5–15 min, the assessment of the colour and smell of the infusions is higher compared with the samples with longer brewing times.

In the case of sensory profiling of flavour and smell (9-point scale), the most distinct herbal flavour (corresponding to the flavour of *Ginkgo biloba*) is characterised by the L20 sample (5.00 points), and the herbal smell—by samples L20 (4.30 points), L15 (4.00 points), and M10, M15 and M20 (5.00 points). The perceptibility of sour flavour is the strongest in L15 (3.80 points) and L20 (3.50 points); however, these assessments do not reduce the overall acceptability of the infusion. The perception of the bitter smell and flavour increases with the increase in the weight of the dried leaves and with the lengthening of the brewing time. The most intense smell is achieved by L10 (2.80 points) and flavour—by L5 (6.30 points) (Figure 7), which are the main factors lowering the overall acceptability of the infusion. The astringency in relation to flavour and smell is most intensely felt in the L60 tests (it scored 3.00 and 3.50 points, respectively). The sweetness indicator is the highest in the case of the X10 sample: both in the case of flavour and smell, it received 1.50 points.

The bitter flavour of infusions is associated with the increased migration of polyphenols occurring during the extension of the brewing time, which results in a more intense experience of this flavour. The substances responsible for this impression mainly include catechin and tannins—e.g., epigallocatechin gallate [41,42]. The tannins and alkaloids are mainly responsible for the tart aftertaste of infusions [43]. Sweetness is noticeable in infusions due to the dominance of carbohydrates in *Ginkgo biloba* leaves; moreover, the drink also contains amino acids that give it a sweet taste [44]. However, the concentration of sweet substances in the leaves is very low, which contributes to their low impact on taste [43].

### 3.5. Analyses of Micro- and Macroelements

Dried *Gingko biloba* leaves used in our own research did not contain heavy metals (Pb, Cd, Ni, Cu) that could adversely affect the functioning of the body. Cr, Fe, K and Mg show increased migration to the infusion with the increasing weight of the sample and the lengthening of the brewing time. The migration of Ca, Na and Mn is the strongest in infusions prepared from a smaller mass of raw material (S). Moreover, in the case of Ca (1.64 g/kg) and N (0.008 mg/kg), the highest concentrations are observed in samples steamed for 15 min. Cr is present in the leaves at 0.48 mg/kg (Table 9). The safe value is 0.05–0.20 mg/kg [44], while in infusions, its concentration has dropped to the level of 0.05–0.08 mg/kg, which is a safe level. Fe had the smallest share (the maximum concentration was found in the M60 sample: 0.08 mg/kg).

Liang et al. (2017) [45] found the presence of lead and cadmium in washed *Ginkgo biloba* leaves, but these amounts were safe for human health. The reference intake value of Cr for an adult is 0.04 mg/kg [46], which makes this raw material a good source of this element. A study by Zhao et al. (2022) [47] on the effect of chromium supplementation on blood glucose and lipid levels in patients with type 2 diabetes showed that this element could, to some extent, reduce the level of glycosylated haemoglobin in patients with carbohydrate metabolism disorders. The content of Mg, Mn, K and Ca in the infusions in own research exceeds the reference intake values [46], which proves that the *Ginkgo biloba* infusion is a good source of them. In our own studies, a higher content of minerals Zn, Fe, Na, Mg, K, Ca and N were found than in dried *Ginkgo biloba* leaves tested in the USA [48].

## 4. Conclusions

The results of our own research confirm the high antioxidant activity and the ability to capture free radicals of the tested infusions prepared from dried *Ginkgo biloba* leaves. The highest activity was shown by infusions obtained from larger weights of dried ginkgo leaves (M and L) brewed for 15 to 60 min; however, the organoleptic evaluation showed that the samples with an average content of dried material (S and M) brewed from 10 to 15 min were the most attractive. According to the assessment of overall acceptability, the M15 trial was the best.

The research showed a very good mineral composition of the raw material and no heavy metals. A relationship between the dose and time of brewing the leaves and the transfer of key minerals to the infusion (especially Cr, Fe, K and Mg) was observed. The only heavy metal found in the leaves was Cr. However, it was present in such small amounts that it did not cause toxicity and could have a health-promoting effect.

A high correlation between the content of bioactive compounds (and antioxidant activity) and the migration of mineral components with increasing weight and lengthening of brewing time was found. This was particularly evident in the case of total polyphenol content and antioxidant activity determined by the TEAC method. To sum up, the most favourable antioxidant activity and, at the same time, attractive sensory values were found in samples M15 and L5. The obtained results suggest that regular consumption of infusions from *Ginkgo biloba* leaves (hence the free radical scavenging potential also confirmed by many studies) may have a protective effect against the more and more frequent civilisation diseases.

## Figures and Tables

**Figure 1 antioxidants-12-01455-f001:**
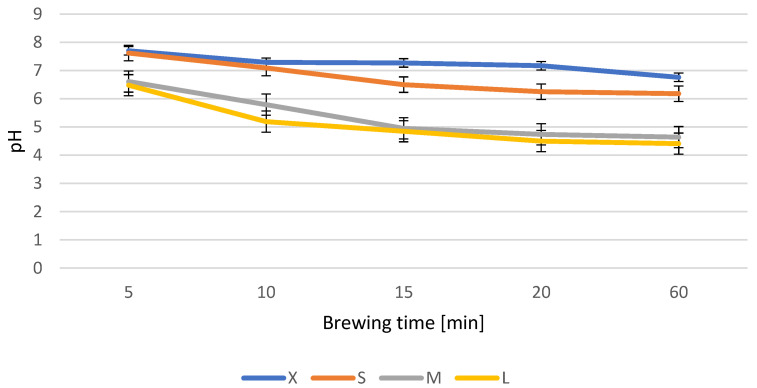
Changes in the pH of various weights in relation to the length of brewing the infusion.

**Figure 2 antioxidants-12-01455-f002:**
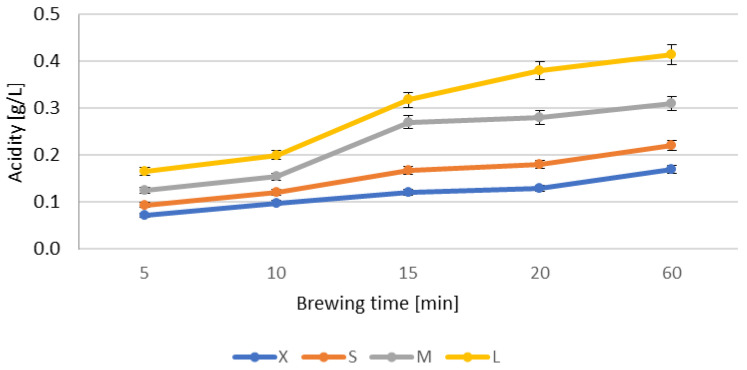
Changes in the acidity [g/L] of various weights in relation to the length of brewing the infusion.

**Figure 3 antioxidants-12-01455-f003:**
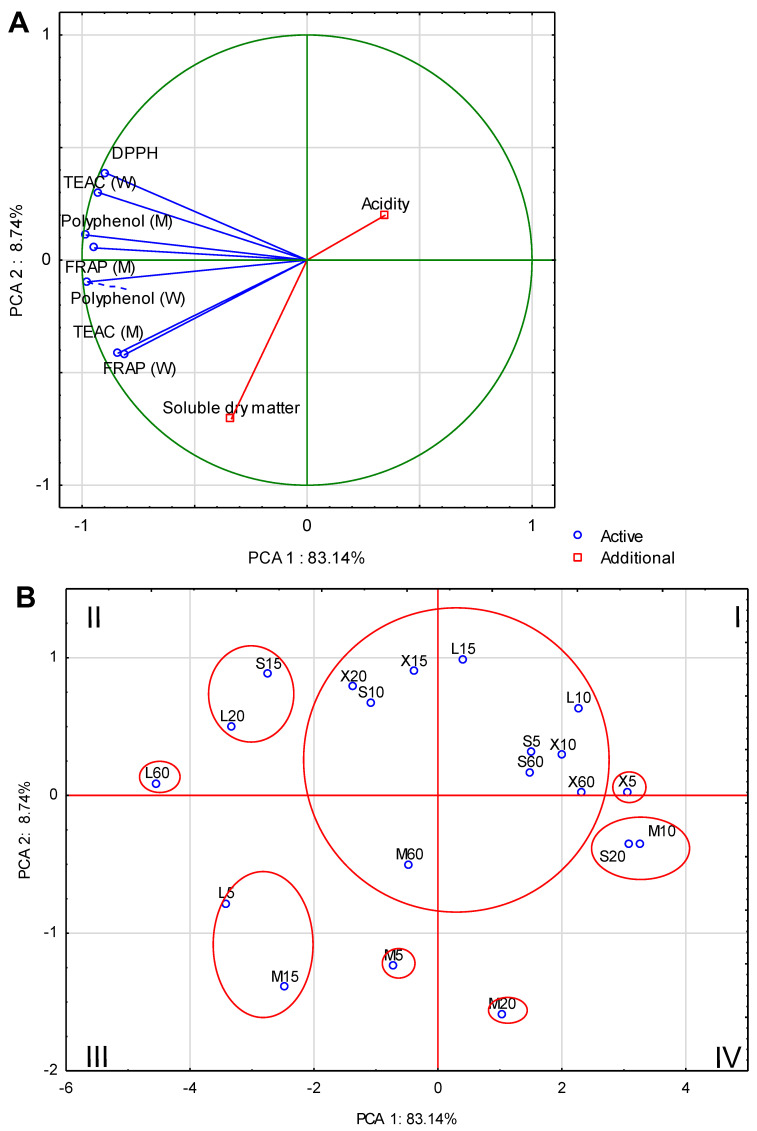
PCA biplot based on the first two principal component axes for antioxidant activity, acidity and soluble dry matter (**A**) and distribution of the 20 test samples based on the first two components derived from principal constituent analysis (**B**).

**Figure 4 antioxidants-12-01455-f004:**
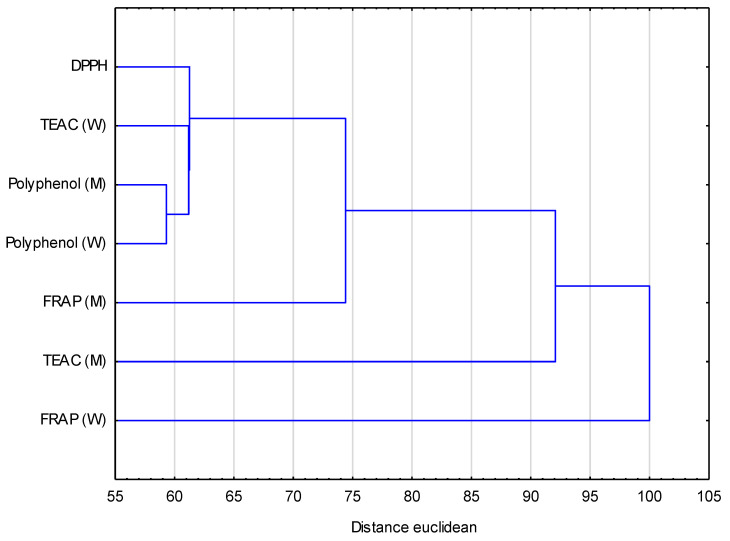
Hierarchical method of agglomeration of clusters of the tested samples in relation to the antioxidant activity (total polyphenol content of the water (W) and methanol (M) fractions, DPPH, FRAP of the water (W) and methanol (M) fractions and TEAC of the water (W) and methanol (M) fractions). Cluster analysis is expressed in Euclidean distance [binding distance/maximum distance × 100].

**Figure 5 antioxidants-12-01455-f005:**
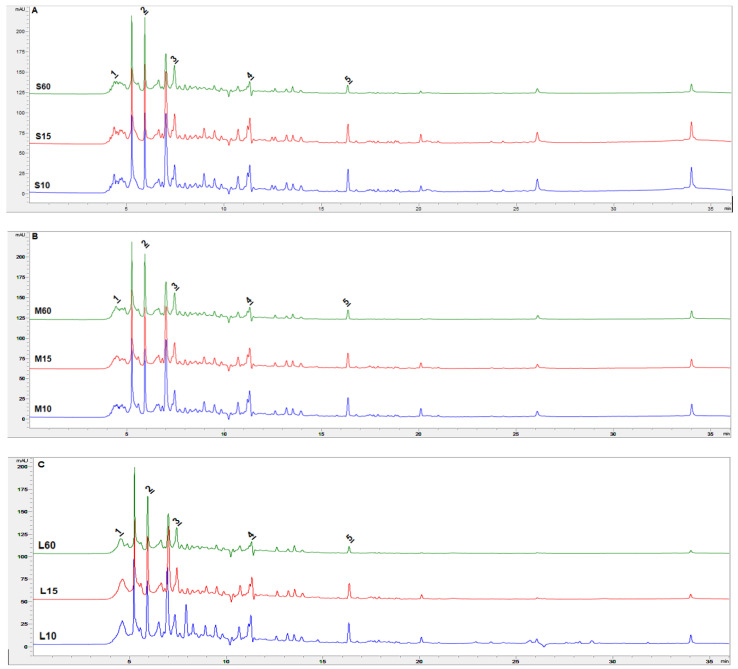
HPLC profile of infusions from dried *Ginkgo biloba* leaves in selected samples ((**A**)—samples with 2.5 g weight—S, brewing time 10, 15 and 60 min; (**B**)—samples with 5.0 g weight—M, brewing time 10, 15 and 60 min; (**C**)—samples with 7.5 g weight—L, brewing time 10, 15 and 60 min). All samples were determined—(1) chlorogenic acid; (2) epicatechin; (3) catechin; (4) coumarin; and (5) apigenin—with detection at 280 nm.

**Figure 6 antioxidants-12-01455-f006:**
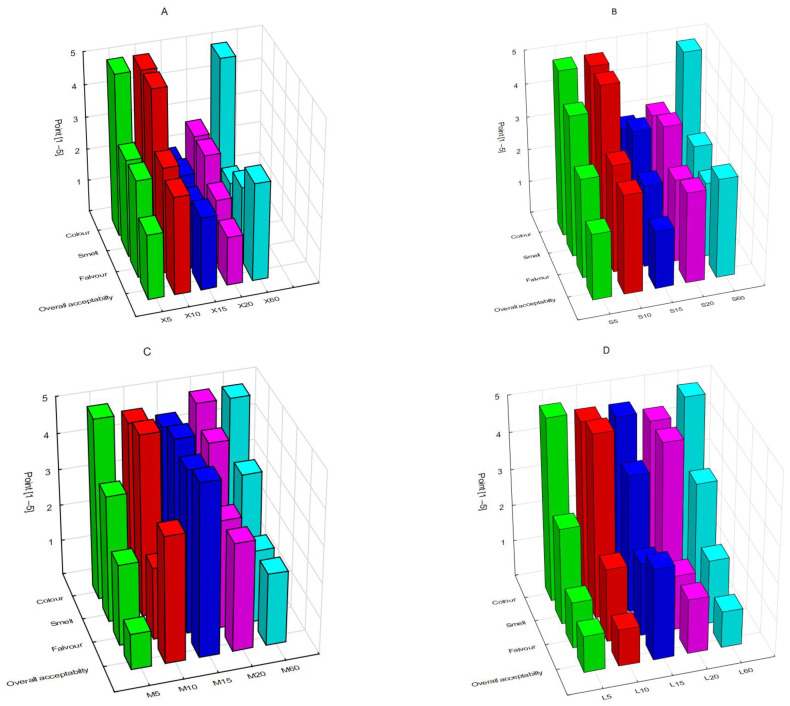
Comparing colour, smell, flavour and overall acceptability of samples X—the smallest weight (**A**), S—small weight (**B**), M—medium weight (**C**) and L—large weight (**D**).

**Figure 7 antioxidants-12-01455-f007:**
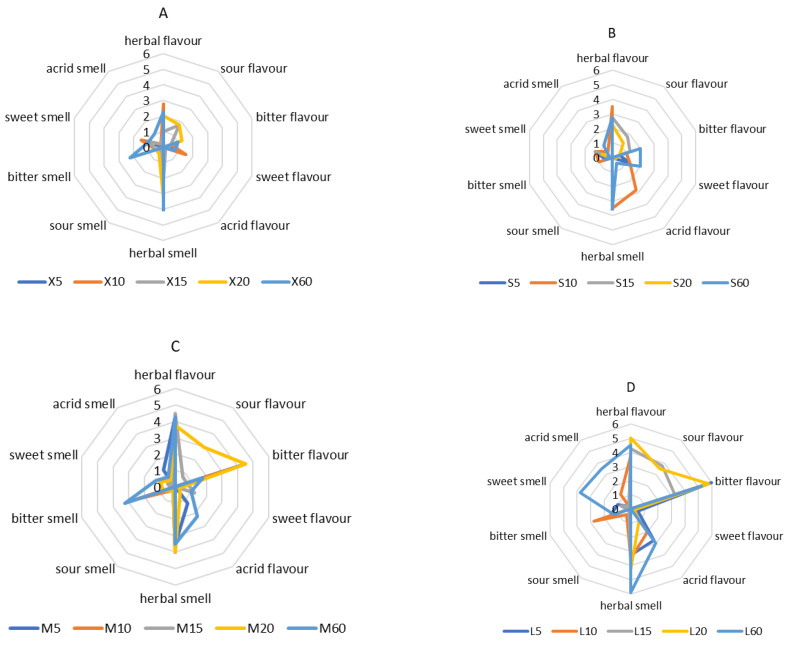
Sensory profile of smell and flavour of samples X—the smallest weight (**A**), S—small weight (**B**), M—medium weight (**C**) and L—large weight (**D**) for attributes such as herbal, acrid, sweet, bitter and sour. For the clarity of the graphs, the scale has been limited to six.

**Table 1 antioxidants-12-01455-t001:** Variants of infusions included in the research.

		Brewing Time [min]				
		5	10	15	20	60
Weight [g] and Symbols	1.25 (X)	X5	X10	X15	X20	X60
	2.50 (S)	S5	S10	S15	S20	S60
	5.00 (M)	M5	M10	M15	M20	M60
	7.50 (L)	L5	L10	L15	L20	L60

**Table 2 antioxidants-12-01455-t002:** Basic characteristics of dry *Ginkgo biloba* leaves.

Parameters	Value
pH	4.09 ± 0.00
Dry weight [%]	92.31 ± 0.13
Ash [%]	0.09 ± 0.01
Fat [%]	5.04 ± 0.36
Protein [%]	11.48 ± 0.17
Carbohydrates [%]	75.70

**Table 3 antioxidants-12-01455-t003:** Total polyphenol content and antioxidant activities in dried *Ginkgo biloba* leaves extracts. Letters in superscript represent significant differences (*p* < 0.05) for the same activity but different acts (in column). The same letter indicates no significant difference (*p* > 0.05).

	Total Polyphenol Content [mg GAE/g]	FRAP [µg TE/g]	DPPH [µg TE/g]	TEAC [µg TE/g]
LWC	85.786 ^a^ ± 0.521	0.322 ^a^ ± 0.004	-	10.751 ^a^ ± 0.209
LWH	77.968 ^a^ ± 0.489	0.277 ^b^ ± 0.008	-	11.416 ^b^ ± 0.104
LM	169.492 ^b^ ± 0.489	6.439 ^c^ ± 0.010	10.852 ± 0.000	17.786 ^c^ ± 0.225

**Table 4 antioxidants-12-01455-t004:** Changes in antioxidant activity in infusions of dried *Ginkgo biloba* leaves. Letters in superscript represent significant differences (*p* < 0.05) for the same activity but different acts, and the same letter indicates no significant difference (*p* > 0.05).

	DPPH [µg TE/mL]	TEAC (M) [µg TE/mL]	TEAC (W) [µg TE/mL]	FRAP (M) [µg TE/mL]	FRAP (W) [µg TE/mL]	Total Polyphenol Content (M) [mg GAE/mL]	Total Polyphenol Content (W) [mg GAE/mL]
X5	0.020 ^a^ ± 0.009	0.408 ^e^ ± 0.003	0.701 ^d^ ± 0.002	0.063 ^b^ ± 0.003	0.045 ^a^ ± 0.002	0.315 ^c^ ± 0.018	0.169 ^a^ ± 0.012
X10	0.038 ^b^ ± 0.008	0.408 ^e^ ± 0.005	0.845 ^g^ ± 0.025	0.313 ^a^ ± 0.003	0.061 ^b^ ± 0.001	0.350 ^d^ ± 0.012	0.214 ^b^ ± 0.000
X15	0.237 ^j^ ± 0.005	0.471 ^f^ ± 0.008	1.309 ^l^ ± 0.002	0.261 ^a^ ± 0.006	0.262 ^k^ ± 0.002	0.716 ^l^ ± 0.005	0.596 ^j^ ± 0.010
X20	0.268 ^k^ ± 0.007	0.521 ^g^ ± 0.002	1.356 ^n^ ± 0.015	0.314 ^a^ ± 0.008	0.349 ^n^ ± 0.006	0.883 ^p^ ± 0.009	0.759 ^n^ ± 0.008
X60	0.056 ^c^ ± 0.007	0.561 ^i^ ± 0.010	0.814 ^f^ ± 0.010	0.120 ^a^ ± 0.006	0.076 ^d^ ± 0.002	0.379 ^e^ ± 0.006	0.222 ^d^ ± 0.007
S5	0.103 ^e^ ± 0.001	0.544 ^h^ ± 0.005	1.033 ^i^ ± 0.020	0.141 ^a^ ± 0.003	0.117 ^g^ ± 0.006	0.494 ^h^ ± 0.010	0.366 ^g^ ± 0.010
S10	0.234 ^j^ ± 0.005	0.696 ^k^ ± 0.020	1.511 ^p^ ± 0.010	0.277 ^a^ ± 0.010	0.289 ^l^ ± 0.001	0.832 ^n^ ± 0.016	0.671 ^k^ ± 0.007
S15	0.300 ^m^ ± 0.005	0.724 ^l^ ± 0.023	1.822 ^r^ ± 0.030	0.392 ^a^ ± 0.015	0.406 ^o^ ± 0.001	1.033 ^q^ ± 0.000	0.907 ^o^ ± 0.013
S20	0.033 ^b^ ± 0.006	0.406 ^d^ ± 0.050	0.272 ^b^ ± 0.022	0.092 ^a^ ± 0.004	0.088 ^e^ ± 0.008	0.316 ^b^ ± 0.000	0.216 ^c^ ± 0.010
S60	0.128 ^f^ ± 0.002	0.471 ^f^ ± 0.080	0.739 ^e^ ± 0.010	0.188 ^a^ ± 0.002	0.198 ^i^ ± 0.005	0.543 ^i^ ± 0.000	0.216 ^c^ ± 0.007
M5	0.098 ^e^ ± 0.002	0.941 ^m^ ± 0.041	1.091 ^k^ ± 0.030	0.270 ^a^ ± 0.000	0.444 ^r^ ± 0.004	0.727 ^m^ ± 0.005	0.745 ^m^ ± 0.008
M10	0.023 ^a^ ± 0.003	0.329 ^b^ ± 0.044	0.179 ^a^ ± 0.017	0.077 ^a^ ± 0.009	0.068 ^c^ ± 0.007	0.291 ^a^ ± 0.026	0.302 ^f^ ± 0.004
M15	0.170 ^h^ ± 0.004	1.334 ^p^ ± 0.025	1.441 ^o^ ± 0.024	0.355 ^a^ ± 0.004	0.532 ^t^ ± 0.003	0.877 ^o^ ± 0.019	0.933 ^p^ ± 0.016
M20	0.075 ^d^ ± 0.004	0.574 ^j^ ± 0.025	0.491 ^c^ ± 0.016	0.152 ^a^ ± 0.002	0.505 ^s^ ± 0.011	0.457 ^g^ ± 0.008	0.495 ^h^ ± 0.005
M60	0.137 ^g^ ± 0.003	0.986 ^n^ ± 0.053	1.083 ^j^ ± 0.016	0.265 ^a^ ± 0.006	0.261 ^j^ ± 0.003	0.709 ^k^ ± 0.011	0.725 ^l^ ± 0.009
L5	0.214 ^i^ ± 0.001	1.573 ^r^ ± 0.051	1.755 ^q^ ± 0.040	0.448 ^a^ ± 0.005	0.418 ^p^ ± 0.003	1.039 ^r^ ± 0.028	0.991 ^q^ ± 0.014
L10	0.078 ^d^ ± 0.008	0.236 ^a^ ± 0.007	0.963 ^h^ ± 0.070	0.120 ^a^ ± 0.003	0.091 ^f^ ± 0.000	0.427 ^f^ ± 0.002	0.262 ^e^ ± 0.010
L15	0.176 ^h^ ± 0.003	0.354 ^c^ ± 0.007	1.333 ^m^ ± 0.015	0.224 ^a^ ± 0.007	0.187 ^h^ ± 0.002	0.648 ^j^ ± 0.016	0.505 ^i^ ± 0.013
L20	0.282 ^l^ ± 0.002	1.198 ^o^ ± 0.007	1.990 ^s^ ± 0.012	0.411 ^a^ ± 0.024	0.339 ^m^ ± 0.000	1.044 ^t^ ± 0.027	0.992 ^r^ ± 0.014
L60	0.302 ^m^ ± 0.001	1.374 ^q^ ± 0.005	2.162 ^t^ ± 0.022	0.530 ^a^ ± 0.015	0.433 ^q^ ± 0.021	1.098 ^s^ ± 0.013	1.317 ^s^ ± 0.001

**Table 5 antioxidants-12-01455-t005:** Changes in the soluble dry matter [%] content of various weights in relation to the length of brewing the infusion. Letters in superscript represent significant differences (*p* < 0.05) for the same activity but different acts. The same letter indicates no significant difference (*p* > 0.05).

	X	S	M	L
5	0.000 ^a^ ± 0.000	0.000 ^a^ ± 0.000	0.100 ^ab^ ± 0.000	0.100 ^a^ ± 0.000
10	0.030 ^ab^ ± 0.001	0.000 ^a^ ± 0.000	0.170 ^b^ ± 0.001	0.130 ^a^ ± 0.002
15	0.370 ^d^ ± 0.000	0.300 ^b^ ± 0.004	0.300 ^c^ ± 0.003	0.370 ^b^ ± 0.000
20	0.570 ^c^ ± 0.002	0.570 ^c^ ± 0.002	0.570 ^d^ ± 0.000	0.570 ^c^ ± 0.000
60	0.000 ^a^ ± 0.000	0.000 ^a^ ± 0.000	0.020 ^a^ ± 0.000	0.500 ^c^ ± 0.001

**Table 6 antioxidants-12-01455-t006:** The value of the correlation coefficient between antioxidant activities.

	DPPH	TEAC (M)	TEAC (W)	FRAP (M)	FRAP (W)	Polyphenol (M)	Polyphenol (W)	Acidity	Soluble Dry Matter
DPPH		0.54	0.90	0.82	0.63	0.94	0.84	−0.27	0.05
TEAC (M)	0.54		0.69	0.79	0.72	0.77	0.85	−0.39	0.59
TEAC (W)	0.90	0.69		0.89	0.59	0.93	0.86	−0.15	0.06
FRAP (M)	0.82	0.79	0.89		0.69	0.90	0.90	−0.31	0.28
FRAP (W)	0.63	0.72	0.59	0.69		0.76	0.82	−0.32	0.53
Polyphenol (M)	0.94	0.77	0.93	0.90	0.76		0.94	−0.37	0.25
Polyphenol (W)	0.84	0.85	0.86	0.90	0.82	0.94		−0.41	0.44
Acidity	−0.27	−0.39	−0.15	−0.31	−0.32	−0.37	−0.41		−0.58
Soluble dry matter	0.05	0.59	0.06	0.28	0.53	0.25	0.44	−0.58	

**Table 7 antioxidants-12-01455-t007:** Percentage of isolated compounds in selected infusions from dried *Ginkgo biloba* leaves.

Compounds	S10	S15	S60	M10	M15	M60	L10	L15	L60
Chlorogenic acid	4.1	4.4	3.4	6.7	5.2	4.5	8.9	8.2	7.6
Epicatechin	8.1	8.9	11.9	10.0	8.8	11.3	8.3	9.4	12.6
Catechin	4.8	7.5	8.3	7.8	7.4	7.9	5.6	6.2	8.8
Coumarin	9.2	9.3	7.8	9.9	8.0	7.1	7.8	7.7	7.4
Apigenin	4.3	3.6	2.3	4.5	3.5	2.7	3.5	2.8	2.1

The relative error does not exceed 3%.

**Table 8 antioxidants-12-01455-t008:** Changes in total flavonoid content in selected infusions of dried *Ginkgo biloba* leaves. Letters in superscript represent significant differences (*p* < 0.05) for the same activity but different acts. The same letter indicates no significant difference (*p* > 0.05).

	S10	S15	S60	M10	M15	M60	L10	L15	L60
TFC (µg QE/mL)	1.66 ^b^ ± 0.05	2.29 ^c^ ± 0.05	2.48 ^d^ ± 0.05	3.46 ^e^ ± 0.02	4.30 ^f^ ± 0.03	5.02 ^a^ ± 0.02	5.05 ^a^ ± 0.02	5.73 ^g^ ± 0.03	5.97 ^h^ ± 0.03

**Table 9 antioxidants-12-01455-t009:** (**a**) Content of microelements in dried *Ginkgo biloba* leaves and selected infusions prepared from it. (**b**) Content of macroelements in dried *Ginkgo biloba* leaves and selected infusions prepared from it.

(a)							
	Pb [mg/kg]	Cd [mg/kg]	Ni [mg/kg]	Cu [mg/kg]	Cr [mg/kg]	Fe [mg/kg]	Mn [mg/kg]
Dry leaves	-	-	-	-	0.480 ± 0.000	5.501 ± 0.024	22.631 ± 0.200
S10	-	-	-	-	0.051 ± 0.020	0.091 ± 0.016	11.394 ± 0.090
S15	-	-	-	-	0.072 ± 0.000	0.103 ± 0.011	15.203 ± 0.130
S60	-	-	-	-	0.054 ± 0.031	0.122 ± 0.040	20.727 ± 0.011
M10	-	-	-	-	0.056 ± 0.031	0.174 ± 0.041	10.432 ± 0.028
M15	-	-	-	-	0.074 ± 0.012	0.160 ± 0.002	10.551 ± 0.140
M60	-	-	-	-	0.082 ± 0.010	0.188 ± 0.020	17.039 ± 0.081
(**b**)							
	**Ca [g/kg]**	**K [g/kg]**	**Mg [g/kg]**	**Na [mg/kg]**	**N [mg/kg]**		
Dry leaves	13.611 ± 0.020	6.521 ± 0.000	1.041 ± 0.000	113.391 ± 0.020	1.812 ± 0.010		
S10	1.158 ± 0.020	3.454 ± 0.014	2.472 ± 0.031	90.612 ± 0.034	0.004 ± 0.000		
S15	1.645 ± 0.042	4.273 ± 0.061	2.217 ± 0.024	58.854 ± 0.021	0.004 ± 0.000		
S60	1.514 ± 0.051	4.834 ± 0.010	3.182 ± 0.026	59.233 ± 0.010	0.004 ± 0.000		
M10	0.974 ± 0.010	3.632 ± 0.000	2.713 ± 0.010	37.132 ± 0.000	0.003 ± 0.020		
M15	1.081 ± 0.010	3.564 ± 0.021	3.181 ± 0.020	40.381 ± 0.022	0.008 ± 0.000		
M60	1.282 ± 0.000	5.677 ± 0.022	4.100 ± 0.000	46.181 ± 0.010	0.004 ± 0.010		

## Data Availability

Not applicable.

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
