# Peer review of "Effect of Brewing Conditions on Antioxidant Properties of Ginkgo biloba Leaves Infusion"

_antioxidants, 2023, doi:10.3390/antiox12071455_

Round 1
Reviewer 1 Report
1. Giving the section “Abbreviation” in the beginning of manuscript
2. Page 2, line 74: Giving the meaning of AOAC
3. Page 2, line 79: sum1?????
4. Page 3, line 99: in g/L or in g/l, unit consistent through the manuscript
5. Page 3, line 103: …. 7,5 % Na2CO3…..=> …. 7.5 % Na2CO3…
6. Page 4, line 121:…..B is FeCl3=>…..B is FeCl3
7. Page 4, line 141: …(10 min., 15 min. and 60 min.) => …(10 min, 15 min and 60 min)
8. Page 9, line 364:…L20 (3,50 points) =>…L20 (3.50 points)
9. Some conclusions in this manuscript are needed to confirm the related findings, and further elucidate the relationship between antioxidant activities and polyphenols. If authors can provide the relationship between antioxidant activities and flavonoids or terpenoids, the manuscript will be more valuable.
Author Response
Dear Reviewer,
Thank you very much for your thorough review of our manuscript and for your valuable suggestions to improve its quality. In accordance with the suggestions, we included all suggested changes and additions in the work, i.e.:
- We added an "Abbreviations" section at the beginning of the manuscript
- We explained the meaning of AOAC
- We fixed typos
- We introduced a consistent unit throughout the manuscript
- We improved the record of chemical compounds
- Improved leaf brewing time record
- We have corrected the score in the sensory evaluation to the one adopted in the journal
- We have added the results of the total flavonoid analysis (chemical method and HPLC)
Thanks again for your time. We believe that the information added to the publication has greatly enriched it.
Reviewer 2 Report
The study is interesting and after obtaining infusions obtained with cold water and hot water using different amounts of dry Ginkgo biloba leaves
These infusions were extracted with water and methanol (to obtain phenolic compounds) and the extracts' antioxidant capacities were determined using various methods.
Other determinations were made, namely of metals.
In the abstract it would be interesting to have some conclusion of the study
The total polyphenol content increased with increasing leaf weight and infusion time, which is expected.
Although there is a variation in antioxidant capacity depending on the technique used, there was also a tendency for it to increase in extracts obtained with methanol, which, from the outset, tells us that only ingestion of the infusion (made with water) will not have the antioxidant capacity of the methanolic extract.. So, this antioxidant capacity can only be used by ingesting extracts obtained with methanol, evaporating this methanol, and then using the extract as a food supplement, for example. And, therefore, its use to increase health-promoting effect cannot be via infusion ingestion.
The only heavy metal found in the leaves was Cr.
The whole study is interesting and gives important information but needs to be validated by studies that demonstrate these antioxidant activities, at least, in vitro. Thus, in my opinion, a determination of intracellular antioxidant activity should be made to complete this study.
Author Response
Dear Reviewer,
Thank you very much for your thorough review of our manuscript and for your valuable suggestions to improve its quality.
During the revision of the article, following the suggestions received from the Reviewers, we made several significant changes to our manuscript. One of the most important is the addition of the results of the total flavonoid analysis (chemical method and HPLC).
The reviewed publication contains some of the results of our research. Currently, infusions are subjected to biotransformation, and the obtained products are analyzed similarly to infusions. We recognize that the analysis allowing the determination of intracellular antioxidant activity is very important in our research. Unfortunately, we are not able to conduct this research alone - we have to enter into cooperation with another research and development unit. This fact significantly extends the time to obtain results. We plan to carry out this assay for both Ginkgo infusions and the final biotransformation product, and we will publish the results of this study in a second article. In this publication, we plan to compile and compare selected parameters obtained in both parts of the study (1. infusions and 2. biotransformation product). The second publication will therefore complement and extend the results obtained so far.
Thanks again for your time. We believe that the information we have already added to the publication has enriched it. However, for reasons beyond our control (documentation formalities), we will be able to carry out some analyzes only in the near future.
Reviewer 3 Report
The manuscript was to mainly determine the antioxidant properties of infusions from dried Ginkgo biloba leaves on the different brewing time. The manuscript is of some interesting, however, it is lack of strong novelty. In the reviewers' opinion, the authors can address some points below to revise it.
1. Please check the English language throughout the manuscript since some parts were not easy to read.
2. Please format the tables and figures in the manuscript.
3. In the introduction, the authors should emphasize the importance of the study.
4. In the methods, please clarify some key experiment parameters, thus other scientists can repeat and validate the experiments. For example, in the section 2.6, please indicate the experiment detail.
5. In the methods, please add chromatographic methods (i.e., LC-MS) to identify and compare the changes of polyphenolic compounds contents during brewing time.
Please check the English language throughout the manuscript since some parts were not easy to read.
Author Response
Dear Reviewer,
Thank you very much for your thorough review of our manuscript and for your valuable suggestions to improve its quality. In accordance with the suggestions, we included all suggested changes and additions in the work, i.e.:
- We checked and corrected the English language throughout the manuscript, using shorter sentences
- We formatted the tables and figures.
- We supplemented the introduction with more detailed information emphasizing the importance of the study.
- We supplemented the methodology with the required details
- We added the results obtained in liquid chromatography analysis and identified some biologically active compounds.
Thanks again for your time. We believe that the information added to the publication has greatly enriched it.
Round 2
Reviewer 2 Report
The authors clarified some aspects and improved the manuscript
Reviewer 3 Report
The manuscript was improved and I recommend to accept the version.